# A visual sense of number emerges from divisive normalization in a simple center-surround convolutional network

**Joonkoo Park[1,2]\*, David E Huber[1]**

[1]Department of Psychological and Brain Sciences, University of Massachusetts Amherst, Amherst, United States; [2]Commonwealth Honors College, University of Massachusetts Amherst, Amherst, United States

**Abstract** Many species of animals exhibit an intuitive sense of number, suggesting a fundamental neural mechanism for representing numerosity in a visual scene. Recent empirical studies demonstrate that early feedforward visual responses are sensitive to numerosity of a dot array but substantially less so to continuous dimensions orthogonal to numerosity, such as size and spacing of the dots. However, the mechanisms that extract numerosity are unknown. Here, we identified the core neurocomputational principles underlying these effects: (1) center-surround contrast filters; (2) at different spatial scales; with (3) divisive normalization across network units. In an untrained computational model, these principles eliminated sensitivity to size and spacing, making numerosity the main determinant of the neuronal response magnitude. Moreover, a model implementation of these principles explained both well-known and relatively novel illusions of numerosity perception across space and time. This supports the conclusion that the neural structures and feedforward processes that encode numerosity naturally produce visual illusions of numerosity. Taken together, these results identify a set of neurocomputational properties that gives rise to the ubiquity of the number sense in the animal kingdom.

\*For correspondence: joonkoo@umass.edu

**Competing interest:** The authors declare that no competing interests exist.

## Editor's evaluation

The current manuscript presents a computational model of numerosity estimation. The model relies on center-surround contrast filters at different spatial scales with divisive normalization between their responses. Using dot arrays as visual stimuli, the summed normalized responses of the filters are sensitive to numerosity and insensitive to the low-level visual features of dot size and spacing. Importantly, the model provides an explanation of various spatial and temporal illusions in visual numerosity perception.

## Introduction

Humans have an intuitive sense of number that allows numerosity estimation without counting (*Dehaene, 2011*). The prevalence of number sense across phylogeny and ontogeny (*Feigenson et al., 2004*) suggests common neural mechanisms that allow the extraction of numerosity information from a visual scene. While earlier empirical work highlighted the parietal cortex for numerosity representation (*Nieder, 2016*), growing evidence suggests that numerosity is processed at a much earlier stage. A recent study, using high-temporal resolution electroencephalography together with a novel stimulus design, demonstrated that early visual cortical activity is uniquely sensitive to the number (abbreviated as $N$) of a dot array in the absence of any behavioral response, but much less so to nonnumerical dimensions that are orthogonal to number (i.e., size and spacing, abbreviated as $Sz$ and

**Figure 1.** Stimulus design and computational methods. (**A**) Properties of magnitude dimensions represented in three orthogonal axes defined by log-scaled number (*N*), size (*Sz*), and spacing (*Sp*) (**Table 1**). (**B**) Schematic illustration of the computational process from a dot-array image to the driving input (i.e., the model without divisive normalization), *D*, of the simulated neurons, versus the normalized response (i.e., the model with divisive normalization), *R*. A bitmap image of a dot array was fed into a convolutional layer with DoG filters in six different sizes (**Equation 1**). The resulting values, after half wave rectification, represented the driving input. Neighborhood weight, defined by $\eta$, was multiplied by the driving input across all the neurons across all the filter sizes, the summation of which served as the normalization factor (see **Equations 2 and 3**). This illustration of $\eta$ is showing the case where *r* is defined by twice the size of the sigma for the DoG kernel. DOG, difference-of-Gaussians.

*Sp*, respectively; see **Figure 1A**; **Park et al., 2016**). Subsequent behavioral and neural studies showed that this early cortical sensitivity to numerosity indicates feedforward activity in visual areas *V1*, *V2*, and *V3* (**Fornaciai et al., 2017**; **Fornaciai and Park, 2021**; **Fornaciai and Park, 2018**). These results suggest that numerosity is a basic currency of perceived magnitude early in the visual stream.

Nevertheless, it is unclear how feedforward neural activity creates a representation of numerosity within these brain regions. Specifically, the view of numerosity as a *discrete number* of items seems incompatible with the primary modes of information processing in the brain, such as firing rates and population codes, which are *continuous*. Indeed, some authors assume that continuous nonnumerical magnitude information is encoded first and integrated to produce the representation of numerosity

(*Dakin et al., 2011*; *Gebuis et al., 2016*; *Leibovich et al., 2017*). In contradiction, however, recent empirical studies demonstrate that the magnitude of visual cortical activity is most sensitive to number and is relatively insensitive to other continuous dimensions such as size and spacing of a dot array (*DeWind et al., 2019*; *Park, 2018*; *Paul et al., 2022*; *Van Rinsveld et al., 2020*).

What explains this insensitivity to spacing and size effects, despite robust sensitivity to number? Previous computational modeling studies offer some hints to this question. The computational model of *Dehaene and Changeux, 1993* explains numerosity detection based on several neurocomputational principles. That model (hereafter D&C) assumes a one-dimensional linear retina (each dot is a line segment), and responses are normalized across dot size via a convolution layer that represents combinations of two attributes: (1) dot size, as captured by difference-of-Gaussians contrast filters of different widths; and (2) location, by centering filters at different positions. In the convolution layer, the filter that matches the size of each dot dominates the neuronal activity at the location of the dot owing to a winner-take-all lateral inhibition process. To indicate numerosity, a summation layer pools the total activity over all the units in the convolution layer. While the D&C model provided a proof of concept for numerosity detection, it has several limitations as outlined in the discussion. Of these, the most notable is that strong winner-take-all in the convolution layer discretizes visual information (e.g., discrete locations and discrete sizes yielding a literal count of dots), which is implausible for early vision. As a result, the output of the model is completely insensitive to anything other than number in all situations, which is inconsistent with empirical data (*Park et al., 2021*).

Recently, several deep-network-based models have been applied to numerosity perception (*Creatore et al., 2021*; *Kim et al., 2021*; *Nasr et al., 2019*; *Stoianov and Zorzi, 2012*; *Testolin et al., 2020*). *Stoianov and Zorzi, 2012* developed a hierarchical generative model of the sensory input (images of object arrays) and demonstrated that after learning to generate its own sensory input, some units in the hidden layer were sensitive to numerosity irrespective of total area while other units were sensitive to total area irrespective of numerosity. This suggests an unsupervised learning mechanism for efficient coding of the sensory data that can extract statistical regularities of the input images. The authors provided some suggestions as to the specific neurocomputational principle(s) underlying the success of this model. For example, the first hidden layer developed center-surround representations of different sizes and the second layer developed a pattern of inhibitory connections to units in the first layer that encoded cumulative area. However, the development of center-surround detectors based on unsupervised learning is a common observation (*Bell and Sejnowski, 1997*), indicating that such results are not unique to displays of dot arrays, and are instead a natural byproduct of learning in the visual system. In a more recent study, *Kim et al., 2021* found that sensitivity and selectivity to numerosity were well captured in a completely untrained convolutional neural network (AlexNet) (*Krizhevsky et al., 2012*), suggesting that a repeated process of convolution and pooling is capable of normalizing continuous dimensions and extracting numerosity information as a statistical regularity of an image. However, these are 'black box' models, and it is not always clear *how* these models work; these models contain many mechanisms, and it is not clear which mechanisms are crucial for producing numerosity-sensitive units.

Rather than applying a complex multilayer learning model, we distill the neurocomputational principles that enable the visual system to be sensitive to numerosity while remaining relatively insensitive to nonnumerical visual features. These principles are simulated in a single-layer model that does not need to be trained. Consistent with prior work, we hypothesize that center-surround contrast filters at different spatial scales play an important role in numerosity perception. In addition to this 'convolution' of the input, most prior proposals entail some form of pooling or normalization (e.g., normalization between center-surround units). This can emerge across layers of visual processing, as often assumed in 'max pooling' layers of a convolutional neural network (*Scherer et al., 2010*), or it can occur within a layer, as in the strong winner-take-all lateral inhibition used in the *Dehaene and Changeux, 1993* model. Furthermore, some models contain both within-layer normalization and between-layer max pooling (*Krizhevsky et al., 2012*). Although the functional form of within-layer normalization is similar to between-layer max pooling, it differs anatomically, placing the normalized response earlier in visual processing. In determining the neural mechanisms that are core to numerosity, we note that a moderate level of within-layer normalization is consistent with 'divisive normalization' (*Carandini and Heeger, 2011*), in which the response of each neuron reflects its driving input divided by the summation of responses from anatomically surrounding neurons (i.e., a normalization pool). This

normalization is not as extreme as winner-take-all normalization and tends to preserve visual precision through graded activation responses. In the case of early vision, the normalization pool is spatially determined by retinotopic positions. Divisive normalization is known to exist throughout the cortex, reflecting the shunting inhibition of inhibitory interneurons that limit neural activation within a patch of cortex (*Carandini and Heeger, 2011*). A wealth of evidence indicates that divisive normalization is ubiquitous across species and brain systems and hence thought to be a fundamental computation of many neural circuits. Thus, any theory of numerosity perception would be remiss not to include the effect of within-layer divisive normalization.

To determine the contribution of divisive normalization to numerosity encoding, we implemented an untrained neural network with versus without divisive normalization as applied to center-surround filters at different spatial scales (e.g., as in *V*1) (*Figure 1B*). The output simulates the summation of synchronized postsynaptic activity of a large population of neurons at a pre-decisional stage, consistent with previous work (*Fornaciai et al., 2017*; *Park et al., 2016*). Our results show that (1) hierarchically organized multiple center-surround filters of varying size make the network insensitive to spacing and that (2) divisive normalization implemented across network units makes the network additionally insensitive to size. Divisive normalization not only occurs over space but also over time (*Huber and O'Reilly, 2003*). Thus, we additionally implemented temporal divisive normalization to test if it explains the contextual effects of numerosity perception (*Burr and Ross, 2008*; *Park et al., 2021*).

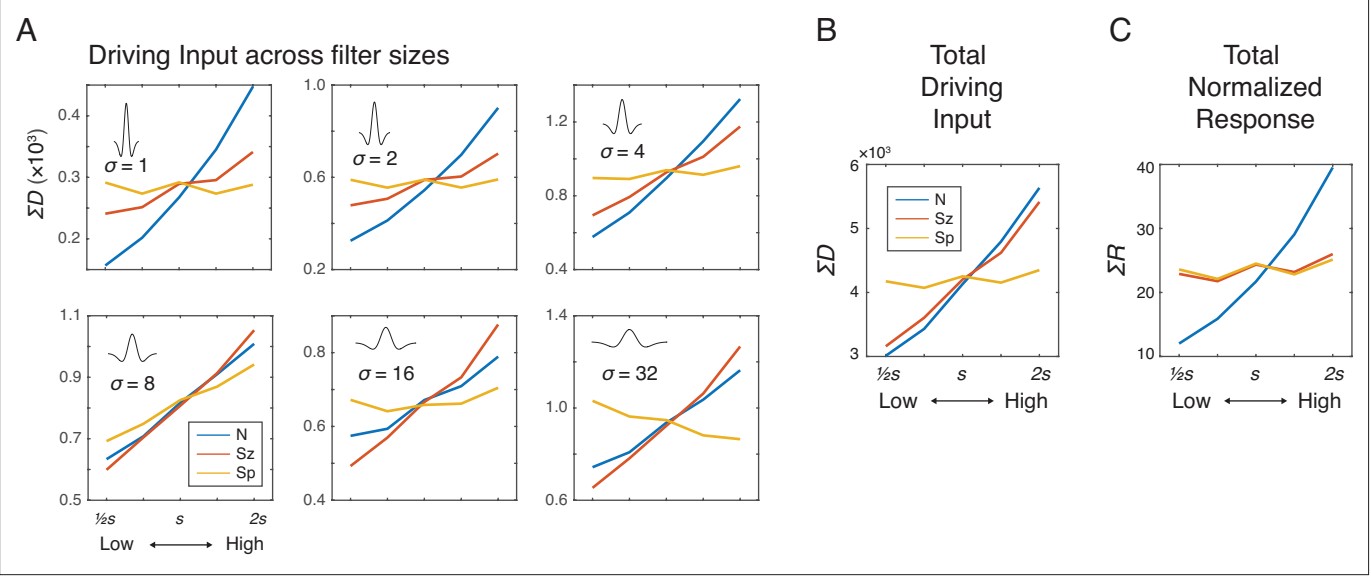

**Figure 2.** Simulation results showing the effects of number (N), size (Sz), and spacing (Sp) on the driving input and normalized response of the network units. (**A**) Summed driving input (Σ D) separately for each of the six filter sizes as a function of N, Sz, and Sp (see Materials and methods for the specific values of s). (**B**) Σ D across all filters is modulated by both number and size but not by spacing. (**C**) Summed normalized response (Σ R) showed a near elimination of the Sz effect leaving only the effect of N. The results were simulated using r=2σ and γ=2, but effects of Sz and Sp were negligible across all the tested model parameters (*Figure 2—figure supplement 2*). The value s on the horizontal axis indicates a median value for each dimension (see Materials and methods).

The online version of this article includes the following figure supplement(s) for figure 2:

**Figure supplement 1.** Additional illustration concerning the driving input.

**Figure supplement 2.** Simulation results showing the effects of number (N), size (Sz), and spacing (Sp) on the normalized response (i.e., the model with divisive normalization) of the network units as a function of neighborhood size (r) and amplification factor (γ).

**Figure supplement 3.** Simulation results from images of densely packed dot arrays with extremely high numerosity.

**Table 1.** Mathematical relationship between various magnitude dimensions.

| Dimension | As a function of $n$, $r_d$, $r_f$ | As a function of $N$, $Sz$, $Sp$ |
|---|---|---|
| Individual area (IA) | $\pi r_d^2$ | $\log(IA) = 1/2 \log(Sz) - 1/2 \log(N)$ |
| Total area (TA) | $n \times \pi r_d^2$ | $\log(TA) = 1/2 \log(Sz) + 1/2 \log(N)$ |
| Field area (FA) | $\pi r_f^2$ | $\log(FA) = 1/2 \log(Sp) + 1/2 \log(N)$ |
| Sparsity (Spar) | $\pi r_f^2/n$ | $\log(Spar) = 1/2 \log(Sp) - 1/2 \log(N)$ |
| Individual perimeter (IP) | $2\pi r_d$ | $\log(IP) = \log(2\sqrt{\pi}) + 1/4 \log(Sz) - 1/4 \log(N)$ |
| Total perimeter (TP) | $n \times 2\pi r_d$ | $\log(TP) = \log(2\sqrt{\pi}) + 1/4 \log(Sz) + 3/4 \log(N)$ |
| Coverage (Cov) | $n \times r_d^2/r_f^2$ | $\log(Cov) = 1/2 \log(Sz) - 1/2 \log(Sp)$ |
| Closeness (Close) | $\pi^2 \times r_d^2 \times r_f^2$ | $\log(Close) = 1/2 \log(Sz) + 1/2 \log(Sp)$ |

Note: n=number; $r_d$=radius of individual dot; $r_f$=radius of the invisible circular field in which the dots are drawn.

## Results

### Center-surround convolution captures total pixel intensities and eliminates the effect of spacing

Images of dot arrays that varied systematically across number, size, and spacing (see Materials and methods) were fed into a convolutional layer with difference-of-Gaussians (DoG) filters in six different sizes. The driving input, $D$, for each filter was the convolution of a DoG with the display image, in other words a weighted sum of local pixel intensities (**Figure 1B**). The summed driving input in each filter size showed different effects as a function of number, size, and spacing (**Figure 2A**), but when the driving input was summed across all filter sizes it was most strongly modulated by both number and size equally but not by spacing (**Figure 2B**), suggesting that the neural activity tracks total area (**TA**; see **Table 1**; **Figure 2—figure supplement 1**). The effect of spacing existed in the fourth and sixth largest filter sizes, largely indicating effects of field area and density, respectively (**Figure 2A**); however, the effects in these two filter sizes were in opposite directions, which made the overall effect very small. These results illustrate that having multiple filter sizes is key to normalizing the spacing dimension. In sum, the driving input of the convolutional layer primarily captured total pixel intensity of the image regardless of the spatial configuration of dots.

### Divisive normalization nearly eliminates the effect of size

We next added divisive normalization to the center-surround model, with different parameter values (neighborhood size and amplification factor) to determine the conditions under which divisive normalization might reduce or eliminate the effect of size and whether it might alter the absence of spacing effects in the driving input. Driving input was normalized by the normalization factor defined by a weighted summation of neighboring neurons and filter sizes (**Equation 2**). The summed normalized responses, $\Sigma R$, were strongly modulated by number but much less so, if any, by size and spacing (**Figure 2C**). The pattern of results was largely consistent across different parameter values for neighborhood size ($r$) and amplification factor ($\gamma$) of the normalization model (**Figure 2—figure supplement 2**); therefore, we chose moderate values of $r$ (=2) and $\gamma$ (=2) for subsequent simulations. As one way to quantify these modulatory effects, a simple linear regression with $\Sigma R$ as the dependent variable with mean-centered values of $N$ as the independent variable (as well as $Sz$ and $Sp$ in separate regression models) was performed. Then, the slope estimate was divided by the intercept estimate, so that these effects could be easily compared across different sets of images (see **Figure 2—figure supplement 3**). This baseline-adjusted regression slope for $N$, $Sz$, and $Sp$ was 0.5771, 0.0646, and 0.0321, respectively. A multiple regression model with summed normalized responses as the dependent measure and the three orthogonal dimensions ($N$, $Sz$, and $Sp$) as the independent variables revealed a much

larger coefficient estimate for *N* (*b*=13.68) than for *Sz* (*b*=1.541) and for *Sp* (*b*=0.7809). In sum, a modest degree of divisive normalization eliminated the effect of size and, at the same time, did not alter the absence of spacing effects.

## Divisive normalization across space explains various visual illusions

Next, we considered if the center-surround model with divisive normalization also explains some of the most well-known visual illusions of numerosity perception. If so, this would support the hypothesis that these visual illusions reflect early visual processing at the level of numerosity encoding, without requiring any downstream processing. In other words, early vision may be the root cause of both numerosity encoding and numerosity visual illusions.

Empirical studies have long shown that irregularly spaced arrays (compared with regularly spaced arrays) and arrays with spatially grouped items (compared with ungrouped items) are all underestimated (*Frith and Frit, 1972*; *Ginsburg, 1976*; *van Oeffelen and Vos, 1982*). These illusions were

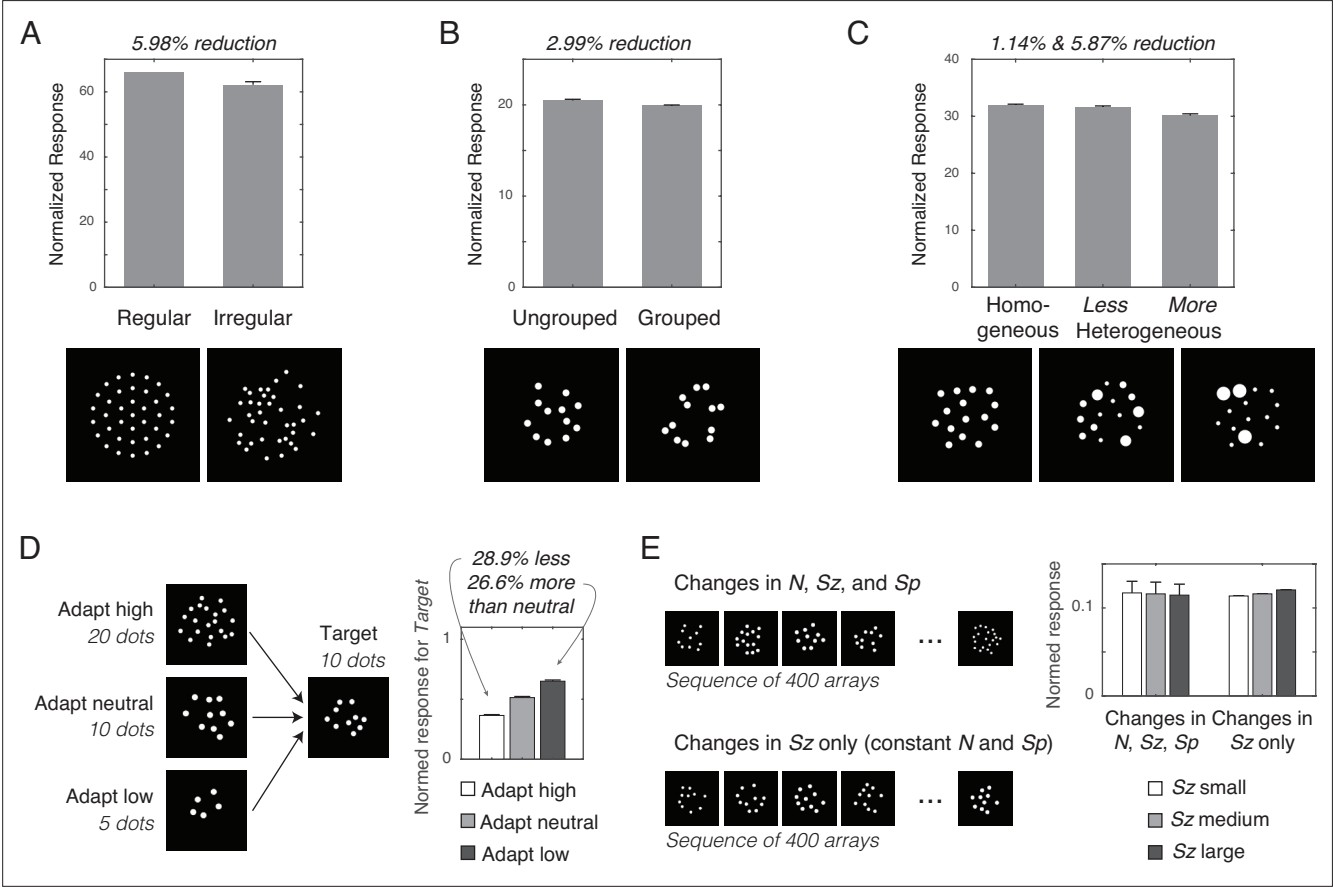

**Figure 3.** Simulation of numerosity illusions. Normalized response of the network units influenced by the (**A**) regularity, (**B**) grouping, and (**C**) heterogeneity of dot arrays, as well as by (**D**) adaptation and (**E**) context. Error bars represent one standard deviation of the normalized response across simulations; however, the error bars in most cases were too small to be visualized. Spatial normalization effects (**A**, **B**, and **C**) were simulated with *r*=2 and *γ*=2. Temporal normalization effects (**D**, **E**) used these same parameters values in combination with *ω*=8 and *δ*=1.

The online version of this article includes the following figure supplement(s) for figure 3:

**Figure supplement 1.** Simulation of visual illusions considering the driving input (i.e., the model without divisive normalization).

**Figure supplement 2.** Effects of single dots.

**Figure supplement 3.** Adaptation effects as a function of model parameters.

**Figure supplement 4.** Adaptation effects along the size dimension.

**Figure supplement 5.** Adaptation effects along the spacing dimension.

**Figure supplement 6.** Context effects as a function of model parameters.

**Figure supplement 7.** Simulation of the connectedness illusion.

indeed captured by the inclusion of divisive normalization. Irregular arrays yielded a 5.98% reduction (Cohen's $d$=4.23) and grouped arrays yielded a 2.99% reduction ($d$=10.02) of normalized response (*Figure 3A–B*). Note that, in the absence of divisive normalization, there was either no effect or an effect in the opposite direction (*Figure 3—figure supplement 1*). The underestimation effects in the normalized response can be explained by greater normalization when neurons with overlapping normalization neighborhoods are activated, with this greater overlap occurring in subregions of the images for irregular or grouped dots. This explanation is functionally similar to one provided by the 'occupancy model' (*Allik and Tuulmets, 1991*), but our results demonstrate that these effects emerge naturally within early visual processing.

A relatively understudied visual illusion is the effect of heterogeneity of dot size on numerosity perception. A recent behavioral study demonstrated that the point of subjective equality was about 5.5% lower in dot arrays with heterogeneous sizes compared with dot arrays with homogeneous sizes (*Lee et al., 2016*). Consistent with this behavioral phenomenon, our simulations revealed that greater heterogeneity leads to greater underestimation (*Figure 3C*). As compared to the homogeneous array, a moderately heterogeneous array (labeled 'less heterogeneous') yielded a 1.14% reduction ($d$=2.43) and the more heterogeneous array yielded a 5.87% reduction ($d$=8.11) in the magnitude of the normalized response. This occurs because the summed normalized response of a single dot saturates as dot area increases (*Figure 3—figure supplement 2*), which interacts with the heterogeneity of the dot array. As heterogeneity is manipulated by making some dots larger and other dots smaller while keeping total area and numerosity constant, this saturating effect makes the overall normalized response smaller as a greater number of dots deviates from the average size (the gains from making some dots larger is not as great as the losses from making some dots smaller). As in the case of other illusions, the same analysis in the absence of divisive normalization fails to produce this illusion (*Figure 3—figure supplement 1*).

## Divisive normalization across time explains numerosity adaptation and context effects

One of the most well-known visual illusions in numerosity perception is the adaptation effect (*Burr and Ross, 2008*). We reasoned that numerosity adaptation might reflect divisive normalization across time, similar to adaptation with light or odor (*Carandini and Heeger, 2011*), which shifts the response curve and produces a contrast aftereffect. Closely related to temporal adaptation, the recently discovered temporal contextual effect of numerosity perception is an amplified neural response to changes in one dimension (e.g., changes in dot size) when observers experience a trial sequence with only changes in that dimension (*Park et al., 2021*). Therefore, we also applied the model with temporal normalization to the context effect.

We modeled temporal divisive normalization for a readout neuron that is driven by the sum of the normalized responses across all units, $\Sigma R$. This summed total response (now referred to as $M$) was temporally normalized ($M^*$) by the recency weighted average of the driving input (*Equation 4*). Temporal normalization shifts the sigmoid response curve horizontally along the dimension of $M$ to maximize the sensitivity of $M^*$ based on the recent history of stimulation. Provided that the constant in the denominator is approximately equal to the current trial's response, the results of spatial normalization reported above would not change by also introducing temporal normalization. Temporal normalization was assessed for cases of a target array of 10 dots after observing an array of 5, 10, or 20 dots with the model parameters of $\omega$=8 and $\delta$=1 (*Figure 3D*) in 32 simulations. Similar to behavioral results (*Aagten-Murphy and Burr, 2016*), the target of 10 dots was underestimated by 28.9% ($d$=18.04) when the adaptor was more numerous than the target and was overestimated by 26.6% ($d$=14.06) when the adaptor was less numerous than the target. This pattern held across all tested model parameters (*Figure 3—figure supplement 3*). It is important to note that the model does not 'know' the number of dots in the adaptor image. Instead, temporal divisive normalization compares the spatially normalized response of the current image to that of the adaptor image and because the spatially normalized response is primarily sensitive to variation in number, there is a contrast effect (e.g., 'adapt high' reduces the response to the current image). Indeed, because the normalized response is less sensitive to variation in size or spacing, no adaptation effect emerges for those variables (*Figure 3—figure supplement 4* and *Figure 3—figure supplement 5*). These results confirm that divisive normalization across space and time naturally produces numerosity adaptation.

Using the same model and parameters of temporal normalization (*Equation 4*), we tested if it can also explain longer-sequence context effects. Studies show that the effect of size is negligible in the context of a trial sequence that varies size, spacing, and number (*Park et al., 2016*), but that the effect of size becomes apparent when number and spacing are held constant while varying only size (*Park et al., 2021*). We simulated each of these contexts: the model saw a total of 400 dot arrays that varied across number, size, and spacing or else it saw 400 dot arrays that differed only in size (*Figure 3E*). A total of 128 simulations were run for each context. In the context where all dimensions varied, the three levels of *Sz* had no linear association with $M^*$; the 95th percentile confidence interval of the ordinary-least-square linear slope of $M^*$ as a function of *Sz* was [–0.0243, 0.0182], which includes 0. In contrast, in the context where only size varied, $M^*$ was positively correlated with *Sz*; slope confidence interval of [0.00315, 0.00359], which excludes 0. This pattern held across all tested model parameters (*Figure 3—figure supplement 6*). This phenomenon can be explained by the adaptive shifting of the sigmoid response curve across trials. In the former case, because recent trials are often of larger or smaller total response as compared to the current trial, the normalization for the current trial is more often pushed to the nonlinear parts of the normalization curve (e.g., closer to ceiling and floor effects). Thus, the temporally normalized response is relatively insensitive to the small effect of size (keeping in mind that the effect of size is made small by spatial divisive normalization). In contrast, when only size varies across trials, the total response of recent trials is more likely to be well-matched to the total response of the current trial. As a result, the small effect of size is magnified in light of this temporal stability.

## Discussion

Despite the ubiquity of number sense across animal species, it was previously unclear how unadulterated perceptual responses produce the full variety of numerosity perception effects. Recent empirical studies demonstrate that feedforward neural activity in early visual areas is uniquely sensitive to the numerosity but much less so, if any, to the dimension of size and spacing, which are continuous nonnumerical dimensions that are orthogonal to numerosity. Despite recent advances showing that numerosity information *can* be extracted from a deep neural network (*Kim et al., 2021*; *Nasr et al.,*

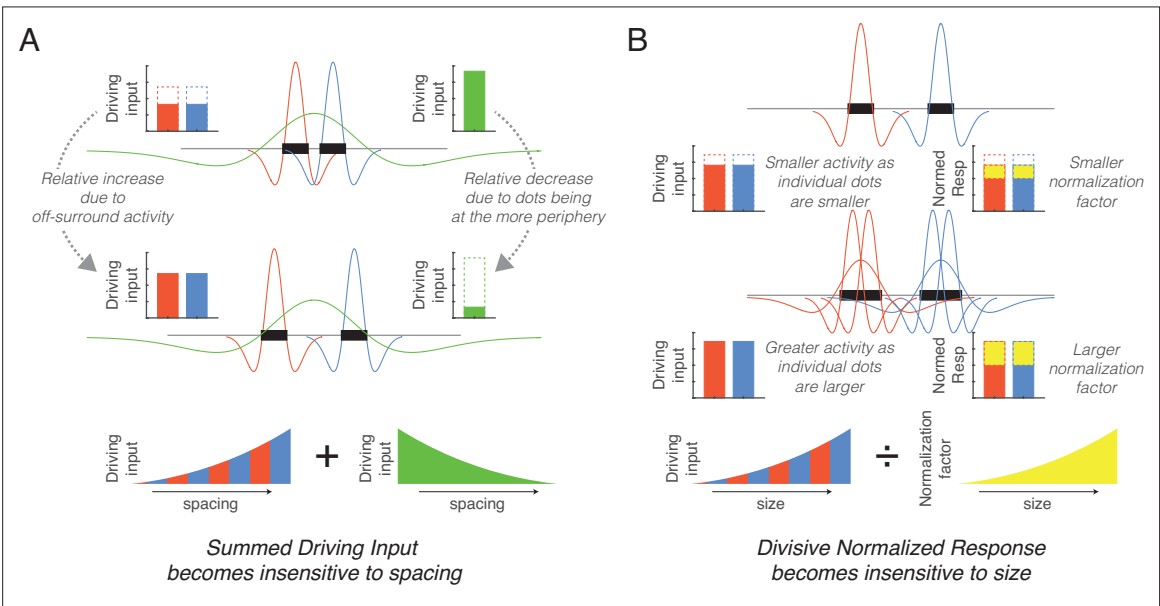

**Figure 4.** Simplified schematics explaining the mechanisms underlying the normalization of size and spacing. (**A**) As spacing increases (from top to middle row) the response of small size center-surround filters increases (red and blue) whereas the response of large size center-surround filters decreases (green), with these effects counteracting each other in the total response. (**B**) As dot size increases (from top to middle row), more filters are involved in responding to the dots thereby increasing the unnormalized response (red and blue), but this results in a greater overlap in the neighborhoods and increases the normalization factor (yellow). These counteracting effects eliminate the size effect.

*2019*; *Stoianov and Zorzi, 2012*), precisely *how* early visual areas normalize the effects of size and spacing was unclear.

The current study identified the key neurocomputational principles involved in this process. First, the implementation of hierarchically organized multiple sizes of center-surround filters effectively normalizes spacing owing to offsetting factors (*Figure 4A*). On the one hand, relatively smaller filters that roughly match or are slightly bigger than each dot produce a greater response when the dots are farther apart because their off-surround receptive fields (RFs) do not overlap. On the other hand, relatively larger filters that cover most of the array produce a greater response when the dots are closer together because stimulation at the center of the on-surround RFs is maximized. When summing these opposing effects, which occur at different center-surround filter sizes, the overall neural activity is relatively invariant to spacing. Second, the implementation of divisive normalization reduces the effect of size by reducing activity at larger filter sizes that have overlapping normalization neighborhoods (*Figure 4B*). More specifically, increase in size produces greater overall unnormalized activity because more filters (e.g., both larger and smaller) are involved in responding to larger dots whereas only smaller filters respond to small dots (*Figure 2B*). However, normalization dampens this increase. Critically, divisive normalization is a within-layer effect, reflecting recurrent inhibition between center-surround filters owing to inhibitory interneurons. Thus, the effect of dot size is eliminated in early visual responses. In sum, contrast filters at different spatial scales and divisive normalization naturally increases sensitivity to the number of items in a visual scene. Because these neurocomputational principles are commonly found in visual animals, this suggests that visual perception of numerosity is a natural, emergent phenomenon.

A key result from the current model is that the summed normalized output of the neuronal activity is sensitive to numerosity but shows little variation with size and spacing. This pattern is consistent with neural studies finding similar results for the summed response of *V1*, *V2*, and *V3* in the absence of any behavioral judgment (*Fornaciai et al., 2017*; *Fornaciai and Park, 2018*; *Paul et al., 2022*). However, this pattern is different than the behavior of prior deep neural network-based models of numerosity perception, which revealed many units in the deep layers that were sensitive to nonnumerical dimensions, along with a few that were numerosity sensitive (or selective). Although the few units that were sensitive to numerosity could explain behavior, the abundance of simulated neurons sensitive to nonnumerical dimensions is inconsistent with population-level neural activity, which fails to show sensitivity to these nonnumerical dimensions in early visual cortex (*DeWind et al., 2019*; *Park, 2018*; *Van Rinsveld et al., 2020*). A key difference between the current model and previous computational models is the inclusion of divisive normalization in the center-surround convolution layer. Unlike prior models, this eliminated the effect of size in the early visual response, without requiring subsequent pooling layers (*Creatore et al., 2021*; *Kim et al., 2021*; *Nasr et al., 2019*; *Stoianov and Zorzi, 2012*; *Testolin et al., 2020*) or a decision making process that compares high versus low spatial frequency responses (*Dakin et al., 2011*).

At first blush, the current model might be considered an extension of *Dehaene and Changeux, 1993*. However, there are four ways in which the current model differs qualitatively from the D&C model. First, the D&C model is one-dimensional, simulating a linear retina, whereas we model a two-dimensional retina feeding into center-surround filters, allowing application to the two-dimensional images used in numerosity experiments (*Figure 1A*). Second, extreme winner-take-all normalization in the convolution layer of the D&C model implausibly limits visual precision by discretizing the visual response. For example, the convolution layer in the D&C model only knows which of 9 possible sizes and 50 possible locations occurred. In contrast, by using divisive normalization in the current model, each dot produces activity at many locations and many filter sizes despite normalization, and a population could be used to determine exact location and size. Third, extreme winner-take-all normalization also eliminates all information other than dot size and location. By using divisive normalization, the current model represents other attributes such as edges and groupings of dots (*Figure 1B*) and these other attributes provide a different explanation of numerosity sensitivity as compared to D&C. For example, the D&C model as applied to the spacing effect between two small dots (*Figure 4A*) would represent the dots as existing discretely at two close locations versus two far locations, with the total summed response being two in either case. In contrast, the current model gives the same total response for a different reason. Although the small filters are less active for closely spaced dots, the closely spaced dots *look like a group* as captured by a larger filter, with this addition for the larger

filter offsetting the loss for the smaller filter. Similarly, as applied to the dot size effect (*Figure 4B*), the D&C model would only represent the larger dots using larger filters. In contrast, the current model represents larger dots with larger filters and with smaller filters that capture the edges of the larger dots, and yet the summed response remains the same in each case owing to divisive normalization (again, there are offsetting factors across different filter sizes). The final difference is that the D&C model does not include temporal normalization, which we show to be critical for explaining adaptation and context effects.

Finally, a recent fMRI study reported that neural activity in *V*1 increases monotonically with numerosity (*Paul et al., 2022*), which is consistent with the current model at a surface level. The authors, however, concluded that this monotonic increase was better explained by aggregate Fourier power than by numerosity. This explanation is qualitatively different than the center-surround and divisive normalization explanation entailed in the current model. While further investigation may be necessary to distinguish these hypotheses, there are two caveats to consider in relation to the conclusions made by *Paul et al., 2022*. First, Fourier power uses spatially unbounded sine waves that have little biological plausibility (unlike center-surround or Gabor filters, which are spatially limited). Second, more critically, the aggregate Fourier power metric used by *Paul et al., 2022* aggregated only up through the first (or an *n*th) harmonic, but the value of the harmonic on the frequency spectrum is dictated by dot size and dot groupings. In other words, the Fourier metric required a priori knowledge about each image, and it is unclear how the visual system could know in advance an appropriate cutoff for a harmonic. Including all frequencies to compute the aggregate Fourier power would likely produce a different conclusion.

Our conclusions are primarily in terms of the qualitative effects of center-surround filtering and divisive normalization, which collectively produce sensitivity to numerosity. However, specific quantitative predictions will change depending on specific model assumptions. For instance, our simulations assumed a distribution of filter sizes that ranged from much smaller to much larger than the presented dots. The responses from filters small enough to capture edges of dots tend to offset the responses from filters large enough to capture local groups of dots, producing relative insensitivity to dot spacing and size (see *Figure 4*). However, there may be extreme cases where this balancing act breaks down. For instance, studies found that when dots are presented in the periphery where RF sizes are larger (*Li et al., 2021*; *Valsecchi et al., 2013*) or if the dots are crowded and hard to individuate (*Anobile et al., 2014*), numerosity perception exhibits different behavioral characteristics. We simulated one extreme by submitting to the model images that contained very small dots (too small to allow edge responses) densely packed in a circular aperture. For this extreme, the summation of normalized responses was still primarily sensitive to number, but that sensitivity was smaller compared to our original simulation, and there was also some moderate sensitivity to size and spacing (*Figure 2—figure supplement 3*). Our simulation also assumed an equal number of small and large center-surround filters although in reality there are likely fewer large filters. This assumption was made out of computational convenience, although we note that similar results would emerge with an unequal distribution of filters if the divisive normalization amplification factor scaled with filter size (e.g., if the larger number of small filters more strongly inhibited each other) or if the neighborhood size of divisive normalization scaled with filter size in a nonlinear manner. By investigating how these assumptions relate to behavior and physiology, future studies may provide additional mechanistic insights into magnitude perception in general.

The success of this model does not necessarily imply that neuronal responses in early visual regions directly determine behavioral responses (see *Fornaciai and Park, 2018*). Prior to behavior, there are many downstream processing steps that incorporate other sources of information, such as response bias and decisional uncertainty. Instead, these results, together with previous electrophysiology results, suggest that normalized response magnitude in early visual regions may be the basic currency from which numerosity judgments are made. Future work should explore the link between the neuronal response layer in the current model and various behavioral judgments. For instance, if decisional uncertainty is modeled by assuming a constant level of decisional noise, regardless of the visual information, then the model will naturally produce Weber's scaling law of just noticeable differences considering that the normalized response follows a log-linear pattern as a function of numerosity (see *Figure 2C*). More complex decisional assumptions could be introduced in an attempt to model the effects of task instructions that are known to bias decisions on magnitude judgment

(*Castaldi et al., 2019*; *Cicchini et al., 2016*). More assumptions about top-down semantic influences may also explain recent coherence illusion results in orientation or color (*DeWind et al., 2020*; *Qu et al., 2022*), for instance, if observers are drawn to focus on a particular feature of the stimulus when comparing two dot arrays.

Another line of possible future work concerns divisive normalization in higher cortical levels involving neurons with more complex RFs. While the current normalization model with center-surround filters successfully explained visual illusions caused by regularity, grouping, and heterogeneity, other numerosity phenomena such as topological invariants and statistical pairing (*He et al., 2015*; *Zhao and Yu, 2016*) may require the action of neurons with RFs that are more complex than center-surround filters. For example, another well-known visual illusion is the effect of connectedness, whereby an array with dots connected pairwise with thin lines is underestimated (by up to 20%) compared to the same array without the lines connected (*Franconeri et al., 2009*). This underestimation effect likely arises from barbell-shaped pairwise groupings of dots, rather than the circularly symmetric groupings of dots that are captured with center-surround filters. Nonetheless, a small magnitude (6%) connectedness illusion emerges with center-surround filters (*Figure 3—figure supplement 7*). Augmenting the current model with a subsequent convolution layer containing oriented line filters and oriented normalization neighborhoods of different sizes might increase the predicted magnitude of the illusion.

In conclusion, our results indicate that divisive normalization in a single convolutional layer with hierarchically organized center-surround filters naturally enhances sensitivity to the discrete number of items in a visual scene by reducing the effects of size and spacing, consistent with recent empirical studies demonstrating direct and rapid encoding of numerosity (*Park et al., 2016*). This account predicts that various well-known numerosity illusions across space and time arise naturally within the same neural responses that encode numerosity, rather than reflecting later stage processes. These results identify the key neurocomputational principles underlying the ubiquity of the number sense in the animal kingdom.

# Materials and methods
## Stimulus sets
### Dot arrays spanning across number, size, and spacing

Inputs to the neural network were visual stimuli of white dot arrays on a black background (200×200 pixels). Dots were homogeneous in size within an array and were drawn within an invisible circular field. Any two dots in an array were at least a diameter apart from edge to edge. The number of dots in an array is referred to as $n$, the radius of each dot is referred to as $r_d$, and the radius of the invisible circular field is referred to as $r_f$. *Table 1* provides mathematical definitions of other nonnumerical dimensions based on these terms.

Following the previously developed framework for systematic dot array construction (*DeWind et al., 2015*; *Park et al., 2016*), stimulus parameters of the dot arrays were distributed systematically within a parameter space defined by three orthogonal dimensions: log-scaled dimensions of number ($N$), size ($Sz$), and spacing ($Sp$) (*Figure 1A*). $N$ simply represents the number of dots. $Sz$ is defined as the dimension that varies with individual area ($IA$) while holding $N$ constant, hence simultaneously varying in total area ($TA$). $Sp$ is defined as the dimension that varies with sparsity ($Spar$) while holding $N$ constant, hence simultaneously varying in field area ($FA$). Log-scaling these dimensions allows $N$, $Sz$, and $Sp$ to be orthogonal to each other and represent all of the nonnumerical dimensions of interest to be represented as a linear combination of those three dimensions (see *Table 1*). Thus, this stimulus construction framework makes is easy to visualize the stimulus parameters and analyze choice behavior or neural data using a linear statistical model. For an implementation of this framework, see the MATLAB code published in the following public repository: https://osf.io/s7xer/.

Across all the dot arrays, number ($n$) ranged between 5 and 20 dots, dot diameter ($2 \times r_d$) ranged between 9 and 18 pixels, field radius ($r_f$) ranged between 45 and 90 pixels, all having five levels in logarithmic scale. $\log(N)$ ranged from 2.322 to 4.322 with the median of 3.322; $\log(Sz)$ ranged from 16.305 to 18.305 with the median of 17.305; $\log(Sp)$ ranged from 19.646 to 21.646 with the median of 20.646. This approach resulted in 35 unique points in the three-dimensional parameter space (see *Figure 1A*). For each of the 35 unique points, a total of 100 dot arrays were randomly constructed for the simulation conducted in this study.

### Dot arrays for testing regularity effects

The 'regular' dot array was constructed following the previous study that first demonstrated the regularity effect (*Ginsburg, 1976*). This array contained 37 dots with $r_d$=3 pixels, one of which at the center of the image and the rest distributed in three concentric circles with the radii of 20, 40, and 60 pixels. The 'irregular' arrays were constructed with the same number of and same sized dots randomly placed with $r_f$=72.5 pixels. This radius for the field area was empirically calculated so that the convex hull of the regular array and the mean convex hull of the irregular arrays were matched. Sixteen irregular arrays were used in the simulation.

### Dot arrays for testing grouping effects

One set of 'ungrouped' dot arrays and another set of 'grouped' dot arrays were constructed. Both ungrouped and grouped arrays contained 12 dots, each of which with $r_d$=4.5 pixels. However, in the ungrouped arrays the dots were randomly dispersed, while in the grouped arrays the dots were spatially grouped in pairs. The edge-to-edge distance between the two dots in each pair was approximately equal to $r_d$. A large number of unique dot arrays were constructed using these criteria for each of the two sets. Then, a subset of unique arrays from each set was chosen so that the convex hull of the arrays between the two sets were numerically matched. A total of 16 grouped and 16 ungrouped arrays entered the simulation.

### Dot arrays for testing heterogeneity effects

Three sets of dot arrays equated in the total area (*TA*) were created. The first set of 'homogeneous' (or zero level of heterogeneity) dot arrays contained $n$=15 with $r_d$=5 pixels within a circular field defined by $r_f$=75 pixels. The second set of 'less heterogeneous' dot arrays contained six dots with $r_d$=3 pixels, six dots with $r_d$=5 pixels, and three dots with $r_d$=7.5 pixels. The last set of 'more heterogeneous' dot arrays contained 12 dots with $r_d$=2.5 pixels and 3 dots with $r_d$=10 pixels. Hence, the total area (*TA*) of all the arrays were approximately identical to each other while the variability of individual area (*IA*) differed across the sets. Rounding errors due to pixelation and anti-aliasing, however, caused differences in the actual cumulative intensity measure of the bitmap images. On average, the cumulative intensity values (0 being black and 1 being white in the bitmap image) were comparable between the three sets of arrays: 1209 in the homogeneous arrays, 1194 in the less heterogeneous arrays, and 1204 in the more heterogeneous arrays. Sixteen arrays in each of the three sets entered the simulation.

## Neural network model with divisive normalization

### Convolution with DOG filters

The model consisted of a convolutional layer with DoG filters of six different sizes, that convolved input values of the aforementioned bitmap images displaying dot arrays. This architecture hence provided a structure for 200×200×6 network units (or simulated neurons) activated by images of dot arrays (*Figure 2*). The DoG filters are formally defined as:

$$\Gamma\left(x,y\right) = I \cdot \left( \frac{1}{2\pi\sigma^2} e^{-\frac{x^2+y^2}{2\sigma^2}} - \frac{1}{2\pi K^2\sigma^2} e^{-\frac{x^2+y^2}{2K^2\sigma^2}} \right), \tag{1}$$

where $I$ is the input image, $\sigma^2$ is the spatial variance of the narrower Gaussian, and $K$ is the scaling factor between the two variances. As recommended by *Marr and Hildreth, 1980*, $K$=1.6 was used to achieve balanced bandwidth and sensitivity of the filters. Considering that the input values range [0 1], the DoG filters were reweighted so that the sum of the positive portion equals to 1 and the sum of the negative portion equals to –1, making the summation across all domains 0. This reweighting ensured that the response is maximized when the input matches the DoG filter regardless of filter size and that the filter produces a response of value 0 if the input is constant across a region regardless of filter size. Finally, the output of this convolution process was followed by half-wave rectification at each simulated neuron (*Heeger, 1991*), where negative responses were replaced by zero. This stipulation sets the 'firing threshold' of the network such that the simulated neurons would not fire if the input does not match its DoG filter.

Six different σ values were used ($\sigma_k$=1, 2, 4, 8, 16, and 32 for filter size $k$, respectively) which together were sensitive enough to represent various visual features of the input images, from the

edge of the smallest dots to the overall landscape of the entire array. The activity of each stimulated neuron, *i*, in filter size *k* following this convolution procedure is referred to as $D_{i,k}$.

## Divisive normalization

Following *Carandini and Heeger, 2011*, the normalization model was defined as:

$$R_{i,k} = \frac{D_{i,k}^{\gamma}}{c+\sum_{j,k}\eta_{(i,j,k)}D_{j,k}^{\gamma}} \ , \tag{2}$$

where distance similarity $\eta_{(i,j)}$ is defined as:

$$\eta_{(i,j,k)} = e^{-\frac{d(i,j)}{r_k}} \tag{3}$$

$D_i$ is the driving input of neuron *i* (i.e., the output of the convolution procedure described above), $d_{(i,j)}$ is the Euclidean distance between neuron *i* and neuron *j* in any filter size, *c* is a constant that prevents division by zero. The denominator minus this constant, which was set to 1, is referred to as the normalization factor. The parameter $r_k$, defined for each filter size, serves to scale between local and global normalization. As $r_k$ gets larger, activities from broader set of neurons constitute the normalization factor. In our model, $r_k$ was defined as a scaling factor of $\sigma_k$ (e.g., $r_k=\sigma_k$, $r_k=2\sigma_k$, or $r_k=4\sigma_k$), so that neurons with larger filter sizes have their normalization factor computed from broader pool of neighboring neurons. The parameter γ determines the degree of amplification of individual inputs and serves to scale between winner-take-all and linear normalization. $R_{i,k}$ represents the normalized response of neuron *i* in filter size *k*.

## Modeling temporal modulation of network units

Normalized responses of simulated neurons were further modeled to capture temporal modulations, with another normalization process this time working across time. First, a read-out neuron was assumed that summed up the normalized responses across all the neurons, $\Sigma R_{i,k}$. This single firing activity, now referred to as *M*, underwent the following temporal normalization process that resulted in the normalized activity $M^*$:

$$M_T^* = \frac{M_T^{\delta}}{c+\sum_{t=1}^{T}\eta_t M_t^{\delta}} \tag{4}$$

The temporal distance $\eta$ is defined as:

$$\eta_t = e^{-\frac{d}{\omega}} \ , \tag{5}$$

where *d* is the distance between time point *t* and *T*. As in *Equations 2 and 3*, *c* is a constant that prevents division by zero, which was set to 1 for convenience. The parameter $\omega$ determines the amount of recent history contributing to the normalization factor, and the parameter δ determines the degree of amplification of $M_t$.

The MATLAB code used to implement the model can be found in the following public repository: https://osf.io/4rwjs/.

## Acknowledgements

The authors thank Dr. Michele Fornaciai for inspiring discussions. This study was supported by the National Science Foundation CAREER Award (BCS1654089) to JP and by the National Institute of Mental Health (RF1MH114277) to DEH.

## Additional information

### Funding

| Funder | Grant reference number | Author |
|---|---|---|
| National Science Foundation | BCS 1654089 | Joonkoo Park |
| National Institute of Mental Health | RF1MH114277 | David E Huber |

The funders had no role in study design, data collection and interpretation, or the decision to submit the work for publication.

### Author contributions

Joonkoo Park, Conceptualization, Software, Formal analysis, Funding acquisition, Methodology, Writing – original draft, Writing – review and editing; David E Huber, Formal analysis, Methodology, Writing – original draft, Writing – review and editing, Conceptualization

### Author ORCIDs

Joonkoo Park ⓘ http://orcid.org/0000-0002-6703-3961
David E Huber ⓘ http://orcid.org/0000-0002-7709-7993

### Decision letter and Author response

Decision letter https://doi.org/10.7554/eLife.80990.sa1
Author response https://doi.org/10.7554/eLife.80990.sa2

## Additional files

### Supplementary files
• MDAR checklist

### Data availability

No empirical datasets were generated during the current study. The source code for the computational model presented in this article are available in the following public repository: https://osf.io/4rwjs/.

The following dataset was generated:

| Author(s) | Year | Dataset title | Dataset URL | Database and Identifier |
|---|---|---|---|---|
| Park J | 2022 | A divisive normalization model of numerosity perception | https://osf.io/4rwjs/ | Open Science Framework, 4rwjs |

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
