## [Editor Report]

The current manuscript presents a computational model of numerosity estimation. The model relies on center-surround contrast filters at different spatial scales with divisive normalization between their responses. Using dot arrays as visual stimuli, the summed normalized responses of the filters are sensitive to numerosity and insensitive to the low-level visual features of dot size and spacing. Importantly, the model provides an explanation of various spatial and temporal illusions in visual numerosity perception.

---

## [Decision Letter]

**Decision letter after peer review:**

Thank you for submitting your article "Neurocomputational principles underlying the number sense" for consideration by *eLife*. Your article has been reviewed by 2 peer reviewers, and the evaluation has been overseen by a Reviewing Editor and Joshua Gold as the Senior Editor. The following individual involved in the review of your submission has agreed to reveal their identity: David Charles Burr (Reviewer #2).

Essential revisions:

There was general enthusiasm for the topic. However, we all agreed that the paper lacked a clear differentiation from earlier work by Dehaene and Changeux (1993). Those authors used a slightly different architecture, but there are many similarities and the present paper did not clearly articulate what novel insights/predictions the current model brings to the table. The authors will need to clarify these novel insights – to the extent possible – and will likely need to make some direct comparisons between the current model and older models. If the authors opt to revise and resubmit, the paper will be sent back to the original two reviewers to evaluate in the context of older work.

*Reviewer #1 (Recommendations for the authors):*

1. "As applied to early vision, this strong winner-take-all mechanism is implausible, as the model would suggest that visual cortex only knows that dots exist, without knowing the size or the location of the dots" (page 3; 3rd paragraph) - this is not true. In the Dehaene and Changeux model, the lateral inhibition in the DoG layer does implement a strong winner-take-all mechanism, however, the size and locations of the dots are still encoded in its output. The locations are topographically encoded, and the locus of activity within the DoG layer encodes (i.e., which filters are activated) encodes dot size. Therefore, the model does not imply that the visual cortex is agnostic to dot size and location. Subsequent stages of the model are indeed primarily concerned with numerosity and not affected by dot size or location, just as is the case for the model in the current manuscript.

2. "Critically, unlike connections between layers, such as with the pooling layers of AlexNet, divisive normalization occurs within a layer (e.g., between center-surround units) through recurrent activation" (page 4; 2nd paragraph) - AlexNet also uses a very similar form of divisive normalization within the convolutional layers (local response normalization). This form of divisive normalization has also been used before in a number of models.

3. The findings in Fig. 3 and Fig. S3 concerning the changes in the model response under different conditions should be backed by appropriate statistical tests.

4. Using the term "driving input" to refer to the rectified output of the convolutional layer is somewhat confusing. Perhaps it would be clearer to use the term "unnormalized response" or something similar.

*Reviewer #2 (Recommendations for the authors):*

I would very much like to see this published and make a few suggestions.

I would drop the paragraph about Paul et al. It is misleading, and not very relevant (as you indeed point out).

They cite the fact that numerosity modulates the pupil response as an example of low-level interaction. I think this is misleading. Although the pupil response is indeed a very basic reflex, it is modulated by high-level processes. It does not imply early computation of numerosity. I think the Collins reference is equally shakey.

Also, it would be useful to test the extremes of a model. For example, we know at very high densities that the rules of numerosity estimation change: what happens to the model there?

Finally, abbreviations should be defined: Sz, Sp, and N are not defined until methods, which makes the text difficult to follow. These should be defined on first use, and probably also in the caption to figure 2.

---

## [Author Response]

Essential revisions:There was general enthusiasm for the topic. However, we all agreed that the paper lacked a clear differentiation from earlier work by Dehaene and Changeux (1993). Those authors used a slightly different architecture, but there are many similarities and the present paper did not clearly articulate what novel insights/predictions the current model brings to the table. The authors will need to clarify these novel insights – to the extent possible – and will likely need to make some direct comparisons between the current model and older models. If the authors opt to revise and resubmit, the paper will be sent back to the original two reviewers to evaluate in the context of older work.

Thanks for the positive appraisal of our paper. We appreciate the reviewers and the editors for constructive feedback. We believe our response, with new analyses and revised text, addresses all the concerns raised

Reviewer #1 (Recommendations for the authors):1. "As applied to early vision, this strong winner-take-all mechanism is implausible, as the model would suggest that visual cortex only knows that dots exist, without knowing the size or the location of the dots" (page 3; 3rd paragraph) - this is not true. In the Dehaene and Changeux model, the lateral inhibition in the DoG layer does implement a strong winner-take-all mechanism, however, the size and locations of the dots are still encoded in its output. The locations are topographically encoded, and the locus of activity within the DoG layer encodes (i.e., which filters are activated) encodes dot size. Therefore, the model does not imply that the visual cortex is agnostic to dot size and location. Subsequent stages of the model are indeed primarily concerned with numerosity and not affected by dot size or location, just as is the case for the model in the current manuscript.

Thank you for pointing out this important issue, which we addressed in the revisions listed above. We were incorrect in stating that the D&C model does not know location or size. Instead, it only knows these properties in a discrete imprecise manner, and it has no information about other attributes such as edges or groupings of dots, which play an important role in the current model’s explanation of numerosity. The quoted text has been removed and the revised paragraph appears in full above.

2. "Critically, unlike connections between layers, such as with the pooling layers of AlexNet, divisive normalization occurs within a layer (e.g., between center-surround units) through recurrent activation" (page 4; 2nd paragraph) - AlexNet also uses a very similar form of divisive normalization within the convolutional layers (local response normalization). This form of divisive normalization has also been used before in a number of models.

This is a good point and we’ve used it to revise the manuscript to make clear the distinction between within-layer normalization (as in this model and as in some aspects of AlexNet) versus between-layer normalization (as in the max pooling layers that are often used in CNNs). These two kinds of normalization/pooling are functionally similar, but make different predictions as to where normalization occurs, and about the strength of normalization. More specifically, “divisive normalization” predicts that sensitivity to number will occur in early visual processing (i.e., within layer) and will be of a moderate strength. The critical paragraph in the introduction now reads:

“Rather than applying a complex multilayer learning model, we distill the neurocomputational principles that enable the visual system to be sensitive to numerosity while remaining relatively insensitive to non-numerical visual features. […] A wealth of evidence indicates that divisive normalization is ubiquitous across species and brain systems and hence thought to be a fundamental computation of many neural circuits. Thus, any theory of numerosity perception would be remiss not to include the effect of within-layer divisive normalization.”

3. The findings in Fig. 3 and Fig. S3 concerning the changes in the model response under different conditions should be backed by appropriate statistical tests.

We believe what the reviewer is asking is whether we've run the model on a sufficient number of different images as to know that our simulation results are not some artifact of specific images. This is a legitimate question. However, we do not believe providing inferential test statistics is useful here because, in such simulations, the statistical significance can always be reached by generating more images to test (provided that there is in fact a real effect that generally occurs for most simulations). The question asked with an inferential test of behavior might be whether the results generalize to a larger population of subjects. With a simulation it would be whether the results generalize to additional not-yet-run simulations, although in this case this is easily achieved by simply running more simulations (with enough simulations, the entire population is sampled and there is no need for inferential statistics). Thus, perhaps a better way to quantify and compare the size of the effects across different conditions is to use an effect size measure, such as Cohen’s d, which tells us how many standard deviations lie between the two means of interest. In addition to the magnitude of the effects (e.g., 5.87% reduction) we reported in the original manuscript, we now report the effect size (e.g., d = 8.11) of all the major comparisons we make in the revised manuscript.

4. Using the term "driving input" to refer to the rectified output of the convolutional layer is somewhat confusing. Perhaps it would be clearer to use the term "unnormalized response" or something similar.

We agree that this is somewhat confusing, but this is the terminology adopted by Carandini & Heeger (2012), and it is commonly used in the literature. Therefore, we are worried that using different terminology might be even more confusing.

Reviewer #2 (Recommendations for the authors):I would very much like to see this published and make a few suggestions.I would drop the paragraph about Paul et al. It is misleading, and not very relevant (as you indeed point out).

We understand the reviewer’s motivation for this suggestion; discussion of Paul et al., (2022) could be seen as a distraction. However, when we shared drafts of the manuscript with colleagues, we faced pushback and criticism for failing to explain how the current model goes beyond or is different from previous computational models and proposals regarding the basis of numerosity perception in early vision. This is also evident from reviewer #1’s request for more information about how the current model differs from the model of Dehaene and Changeux (1993). After addressing reviewer 1’s comments, the revised manuscript now contains three paragraphs in Discussion (p. 11-12) that provide detailed explanations for how the current model differs from previous well-known models of numerosity perception and discussion of Paul et al., no longer seems like a distraction within the context of this comparison to other models. We felt it important to retain the discussion of Paul et al., because we also received request of comparison to that study. More specifically, we were asked what makes our proposal a reasonable model when one of the most recent findings indicates that response in early visual processing reflects spatial frequency analysis, not numerosity (Paul et al., 2022). We suspect that readers of our study will ask the same question and so we would like to retain some version of this paragraph to address this issue.

If the reviewer and editor still think that this paragraph is irrelevant, we would be happy to take specific recommendations to achieve our goal in other ways (i.e., other ways to differentiate our work from previous models, including the “spatial frequency” model proposed in Paul et al.).

They cite the fact that numerosity modulates the pupil response as an example of low-level interaction. I think this is misleading. Although the pupil response is indeed a very basic reflex, it is modulated by high-level processes. It does not imply early computation of numerosity. I think the Collins reference is equally shakey.

We agree that this was somewhat misleading/inaccurate and we have removed this part in the discussion.

Also, it would be useful to test the extremes of a model. For example, we know at very high densities that the rules of numerosity estimation change: what happens to the model there?

This is a good question. We now report this simulation in Figure S9, as mentioned in the discussion with the following text:

“For instance, our simulations assumed a distribution of filter sizes that ranged from much smaller to much larger than the presented dots. The responses from filters small enough to capture edges of dots tends to offset the responses from filters large enough to capture local groups of dots, producing relative insensitivity to dot spacing and size (see Figure 4). However, there may be extreme cases where this balancing act breaks down. For instance, studies found that when dots are presented in the periphery where receptive field sizes are larger (Li et al., 2021; Valsecchi et al., 2013) or if the dots are crowded and hard to individuate (Anobile et al., 2014), numerosity perception exhibits different behavioral characteristics. We simulated one extreme by submitting to the model images that contained very small dots (too small to allow edge responses) densely packed in a circular aperture. For this extreme, the summation of normalized responses was still primarily sensitive to number, but that sensitivity was smaller compared to our original simulation, and there was also some moderate sensitivity to size and spacing (Figure S9).”

In order to compare the effects of *N*, *Sz*, and *Sp* from large numerosities (N = 90 to 360) with the same effects from the original images (N = 5 to 20), regression slope adjusted by the baseline (i.e., slope divided by the intercept from linear regression) was computed. This adjusted slope allows comparison of relative change across the two different sets of images. The caption to Figure S9 reads:

“Figure S9. Simulation results from images of densely packed dot arrays with extremely high numerosity. (A) The dots arrays were systematically constructed ranging equally across the dimensions of *N*, *Sz*, and *Sp*, which was achieved by using the following parameters: number (*n*) = from 90 to 360, dot radius (*r_d_*) = from 1 to 2 pixels, field radius (*r_f_*) = from 45 to 90 pixels. For each point in the 2×2×2 parameters space, 16 unique arrays were created. (B) Examples of dot array images are shown. These images were submitted to the current computational model with the same parameters used in our original analysis (r = 2σ and γ = 2). (C) Summed driving input (ΣD) was modulated primarily by *N* and *Sz*. Summed normalized response (ΣR) was most modulated by *N* but also by *Sz* and *Sp* to some degree. The slope of the linear fit to *N*, *Sz*, and *Sp* adjusted by the baseline (the slope estimate divided by the intercept estimate in the simple regression) was 0.4086, 0.1958, and 0.1488, respectively. Note that this adjusted slope allows comparison of relative change in the response driven by *N*, *Sz*, and *Sp*, despite differences in the overall activity across different sets of images. In our original simulation, the adjusted slopes for *N*, *Sz*, and *Sp* were 0.5771, 0.0646, and 0.0321, respectively. Thus, the same computational network when representing much more densely packed dot arrays seems to show relatively decreased sensitivity to numerosity. These results indicate that neural sensitivity to various magnitude dimensions and the degree of that sensitivity differ based on the assumptions about the distribution of filters and filter sizes.”

Finally, abbreviations should be defined: Sz, Sp, and N are not defined until methods, which makes the text difficult to follow. These should be defined on first use, and probably also in the caption to figure 2.

Done. Thanks for the suggestion.